# Assessment of Hydrogen Energy Industry Chain Based on Hydrogen Production Methods, Storage, and Utilization

Zenon Ziobrowski and Adam Rotkegel *

Institute of Chemical Engineering Polish Academy of Sciences, 44-100 Gliwice, Poland; zenz@iich.gliwice.pl
* Correspondence: arot@iich.gliwice.pl

**Abstract:** To reach climate neutrality by 2050, a goal that the European Union set itself, it is necessary to change and modify the whole EU's energy system through deep decarbonization and reduction of greenhouse-gas emissions. The study presents a current insight into the global energy-transition pathway based on the hydrogen energy industry chain. The paper provides a critical analysis of the role of clean hydrogen based on renewable energy sources (green hydrogen) and fossil-fuels-based hydrogen (blue hydrogen) in the development of a new hydrogen-based economy and the reduction of greenhouse-gas emissions. The actual status, costs, future directions, and recommendations for low-carbon hydrogen development and commercial deployment are addressed. Additionally, the integration of hydrogen production with CCUS technologies is presented.

**Keywords:** hydrogen; decarbonization; clean energy; CCUS

## 1. Introduction

The European Union set itself a goal to reach climate neutrality by the year 2050 through deep decarbonization and higher reduction of greenhouse-gas emissions [1]. To achieve climate neutrality by the year 2050, it is necessary to transform and modify the EU's energy system. It is estimated that the EU's energy system is accountable for about 75% of the EU's greenhouse-gas emissions [2].

The EU Strategy for Energy System Integration [3], the report published by the EU Commission in 2020, provides assumptions and objectives for the energy transition. The current EU energy-consumption model for different sectors of the economy, with particular value chains, infrastructure development, planning procedures, and modes of operations, is not cost efficient and will not allow for obtaining climate neutrality by 2050. New innovative solutions and links between all of the economy's sectors must be created and integrated into the new energy system. This new integrated system should be considered as a whole, connecting different energy-infrastructure facilities with different energy carriers and different energy-consumption sectors.

In this integrated energy system, hydrogen plays a primary role as an energy source and storage for different energy flows. The EU Hydrogen Strategy [4,5] conveys how to take advantage of hydrogen's potential through proper investments, relevant regulations, market development, research, and innovations.

The main preference of the EU Hydrogen Strategy is the development of clean hydrogen (green hydrogen) using renewable energy. Currently, hydrogen production methods based on thermo-chemical and electrolysis processes have a promising future, especially when integrated with a Generation IV nuclear power reactor [6,7]. However, in the transition period, hydrogen based on fossil fuels (blue hydrogen) will be also used to decrease emissions and develop a manageable market.

The EU Hydrogen Strategy presents a three-step plan [4], starting with six gigawatts of electrolyser capacity to produce one million tons of green hydrogen up to 2024. By 2030, the EU plans for the installation of 40 gigawatts of electrolysers capacity to produce

up to 10 million tons of green hydrogen. In the third step, in the years 2030 to 2050, the deployment of mature renewable hydrogen technologies at a large scale across all EU economy sectors will be realized.

According to these plans, the constructed electrolysers will be used for the production of renewable green hydrogen; then, local hotspots will be connected for end users as so-called "hydrogen valleys" joined into a large European hydrogen infrastructure. Lastly, mature clean hydrogen technologies will be deployed and utilized at a large scale [3].

To support this strategy, the EU Commission will introduce appropriate policy, common standards, energy legislation, support investments, logistical networks, and the necessary infrastructure in line with the EU's sustainable development.

Hydrogen has received growing worldwide attention as an exclusive clean-energy solution with many potential applications in the industry, power, and transportation sectors as an energy carrier, storage media, and feedstock. Hydrogen is a carbon-free carrier; when used, it does not emit any pollution and is considered an important solution for the decarbonization of different economic sectors, even those with hard-to-abate carbon emissions. Thus, hydrogen's role is essential for the EU's commitment to achieve carbon neutrality by 2050. However, hydrogen is a leak-prone and indirect greenhouse gas, with a global warming potential GWP of 5.8 over a 100-year time horizon. To be an effective climate solution, hydrogen must be produced cleanly and used wisely [8].

The necessity to reduce greenhouse emissions, together with technological developments in renewable energy, is an additional reason for hydrogen to be a priority in the clean-energy transition process. In the EU strategy published in 2018 [9], the projected part of hydrogen in Europe's energy is to grow from 2% [10] to 13–14% by 2050 [11].

The Commission's recovery plan [12] lays stress on clean hydrogen as an essential issue in the process of the energy transition. The planned investments in electrolysers, up to 2030, range between EUR 24 and EUR 42 billion. EUR 220–340 billion will be required to scale up and connect renewable energy production to the electrolysers. Modernization of existing power plants by coupling with CCUS technologies would require around EUR 11 billion. It is estimated that the storage, transport, and distribution of hydrogen will also need EUR 65 billion in investments [11,12]. Generally, the European investments by 2050 in renewable green hydrogen are about EUR 180–470 billion and, for low-carbon fossil-based blue hydrogen, EUR 3–18 billion [13]. As predicted, clean hydrogen may meet 24% of world's energy requirements by 2050 [14].

The hydrogen energy industry chain contains hydrogen production, storage, transportation, and utilization. Meaningful research on the hydrogen energy chain is still required if hydrogen is to become a main component of the energy market [14].

This study presents a current insight into the global energy-transition pathway for sustainable development by means of hydrogen energy. Comprehensive information is provided on hydrogen production, storage, transportation infrastructure reliability performance and stationary applications. The novel technological directions and developments in hydrogen production, storage and utilization are discussed.

The paper introduces the actual status, possibilities, and challenges conveyed by the hydrogen economy and its development. The potentials of hydrogen production, storage, and distribution methods are described and categorized together with various end users based on the energy sources, available feedstock, and hydrogen utilization. Future directions and recommendations for low-carbon hydrogen development are analyzed.

This study also discusses the coupling and integration of hydrogen production from fossil fuels with CCUS technologies and presents recent efforts toward renewable hydrogen production and its important role in the decarbonization of different economic sectors to reach carbon neutrality by 2050.

## 2. Hydrogen's Role in Global Energy Supply

In Table 1, the global share of energy supply in 2019 is presented. As can be seen, fossil fuels have the largest share, reaching 80.9%, followed by renewable sources with a share of 14.1% of the energy supply. The nuclear energy share of 5.0% is the smallest one [15].

**Table 1.** Share of global energy supply in 2019 [15].

| Energy Supply | | Global Market Share | |
|---|---|---|---|
| Fossil fuels | Oil | 30.9% | 80.9% |
| | Coal | 26.8% | |
| | Natural gas | 23.2% | |
| Nuclear | | 5.0% | |
| Renewables | Biofuels | 9.4% | 14.1% |
| | Hydro | 2.5% | |
| | Solar, wind, waves etc. | 2.2% | |

In Figure 1, the global share of energy supply, electricity generation, and $CO_2$ emissions for various energy sources in Canada are compared [16].

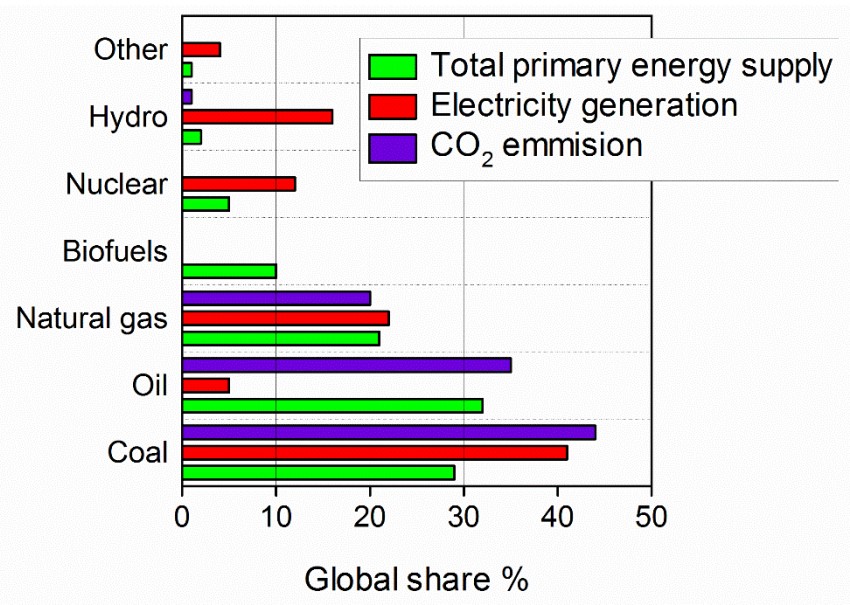

**Figure 1.** Global share of energy supply, electricity generation, and $CO_2$ emissions for various energy sources in Canada.

It can be seen that conventional energy sources, such as oil, coal, and natural gas, possess the largest share of $CO_2$ emissions, electricity generation, and primary energy supply.

The main drawback of consuming conventional energy sources is the growing emission of greenhouse gases and increasing global warming. These environmental problems and extreme challenges are related to the depletion of natural resources and lead to the research of alternative energy sources, new energy carriers, and storage media. Hydrogen has been recognized as an exclusive carbon-free energy solution with many potential applications. Moreover, the existing infrastructure used for storage and transportation in the case of other chemical fuels can also be used for the transport and storage of hydrogen.

In Figure 2, the comparison of hydrogen's lower (red) and higher (blue) heating values (LHV and HHV) [MJ/kg] with conventional fuels is presented. The data show that the hydrogen heating values are significantly higher in comparison with other fuels [17].

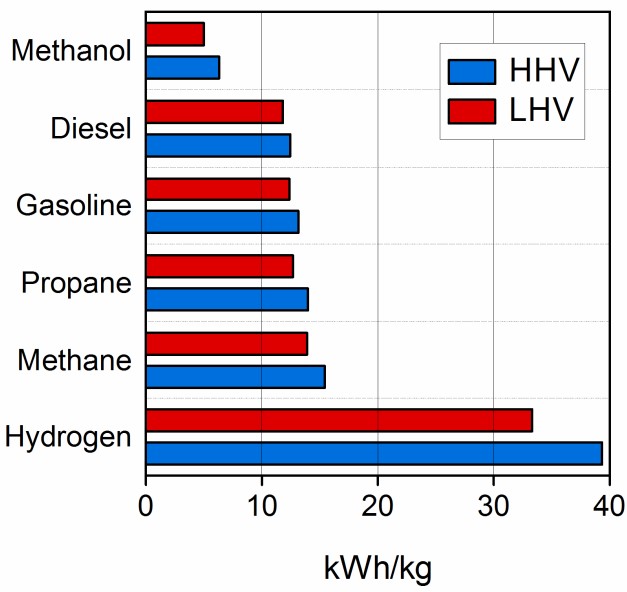

**Figure 2.** Heating values of hydrogen and other fuels.

Liquid fuels represent portable energy carriers which are convenient and attractive applications for different energy production methods. In Table 2, a comparison of the energy densities of hydrogen, conventional liquid fuels and batteries is presented [18].

**Table 2.** Energy densities for hydrogen, liquid fuels, and batteries.

| Fuel | Gravimetric Energy Density kWh/kg | Volumetric Energy Density kWh/dm$^3$ |
|---|---|---|
| Liquid hydrogen (at −253 °C) | | 2.359 |
| Hydrogen (35 MPa) | 33.3 | 0.767 |
| Hydrogen (70 Mpa) | | 1.265 |
| Hydrogen at 0.1 Mpa | | 0.003 |
| Gasoline | 12.3 | 9.06 |
| Diesel | 12.7 | 10.70 |
| Propane | 12.9 | 7.49 |
| Methanol | 5.6 | 4.47 |
| Li-ion battery | 0.15–0.21 | 0.45 |

Hydrogen is a highly energetic fuel, with 33 kWh/kg. In comparison with Tesla battery storage 250–260 Wh/kg, hydrogen gravimetric energy density is about 126 times higher; however, its volumetric energy density is only 3 kWh/m$^3$ at 1 bar and 20 °C. Although the hydrogen gravimetric energy density is very high, its liquid density is only 70.85 kg/m$^3$ (at −253 °C), and its volumetric energy density is 2–5 times lower in comparison with conventional liquid fuels, Table 2.

Compared to gasoline, a hydrogen fuel tank should have 4–5 times higher volume and 10 times higher mass. Liquid hydrogen is an energetically ineffective option because of the low required storage temperature (about −253 °C). Almost 30–40% of its combustion heat is lost in the liquefaction process.

Additionally, hydrogen's ability to penetrate and diffuse through various materials causes some problems with the storage of compressed hydrogen. For this reason, pressurized hydrogen should be stored in tanks capable of withstanding the pressure in the range from 17 MPa to 70 MPa. When weight matters, specially designed tanks made of carbon fibre coated with aluminium or special polymers are used instead of steel.

## 3. The Hydrogen Energy Industry Chain

Building and developing a carbon-neutral hydrogen economy will require a full hydrogen energy chain approach. Renewable and low-carbon sources should be defined. The hydrogen production efficiency, relevant infrastructure, storage, supply chains, and end users should be determined. The hydrogen market needs to be created by determining and estimating affordable costs for clean technologies and energy input from renewable sources.

### 3.1. Hydrogen Production and Utilization

Different conventional and renewable energy sources, technologies, and processes can be used for hydrogen production, which results in different costs, material requirements, and emissions. Depending on the energy source, hydrogen can be produced from conventional energy sources, such as natural gas, coal, oil, and nuclear power [19–21], or from renewable energy sources, such as solar [22], wind [23], biomass [24], geothermal [25], hydro [26], and ocean thermal energy conversion (OTEC) [27–29].

A comparison of hydrogen production methods is shown in Figure 3 for various energy sources and technologies [17].

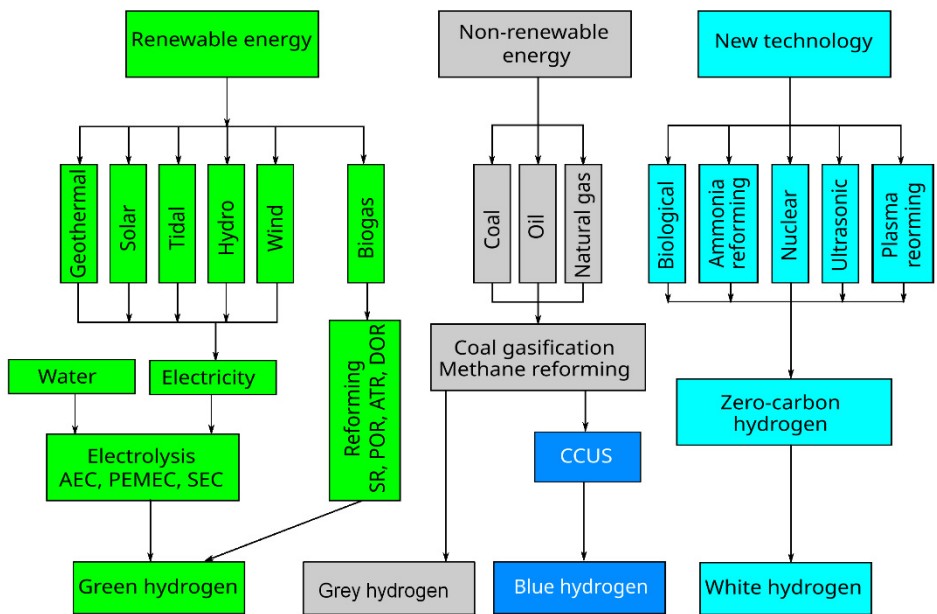

**Figure 3.** Hydrogen production methods based on various energy sources and technologies.

Currently, hydrogen production is mainly based on nonrenewable resources, such as coal, oil, and natural gas [30]. About 95% of the produced hydrogen is based on fossil fuels, whereas hydrogen production from water using electricity and from biomass represents only 4% and 1%, respectively [31]. Hydrogen production is generally classified into three categories depending on conventional or renewable energy sources and other production methods, Figure 3. Hydrogen from renewable energy sources produced through a water electrolysis process is called green because, during the production process, it does not emit any gases. Hydrogen based on fossil fuels and produced commonly by steam methane reforming (SMR) is called grey because it generates a large amount of $CO_2$ emissions and causes environmental problems. Coupling hydrogen production from coal and natural gas with the CCUS processes reduces or eliminates $CO_2$ emissions and thus obtained hydrogen is called blue or low-carbon hydrogen. Green hydrogen is produced based on an electrolysis process powered by solar energy, wind energy, hydro energy, ocean thermal energy conversion, or biomass gasification. Black hydrogen is produced by coal gasification. A large amount of emissions is produced in this process and released into the atmosphere. Hydrogen produced by other methods or new technologies, in small amounts, includes white/purple/aqua hydrogen based on the natural form of $H_2$ in underground deposits, in

oceans and the air, on nuclear power, and new technologies [17,21]. Aqua hydrogen is a new technology for extracting hydrogen from oil sands. By injecting oxygen into underground oil sands, a partial oxidation reaction occurs, producing CO water and $H_2$ and releasing heat. When the temperature rises above 350 °C, water is split into hydrogen and oxygen. Hydrogen is then extracted from the subsurface through permeable membranes, leaving $CO_2$ and other unwanted gases in the reservoir. The aqua hydrogen technology is highly efficient; it does not emit $CO_2$. It is a revolutionary technology promoting zero-carbon hydrogen technology in the case of hydrogen production from fossil fuels. However, it is now in the early phase of development [21,32].

As can be seen in Figure 4, the conversion of fossil fuels is currently the most used method worldwide for hydrogen production (96% of world hydrogen production) [33–35]. Natural gas reforming, methane partial oxidation, and coal gasification are the main sources of hydrogen production.

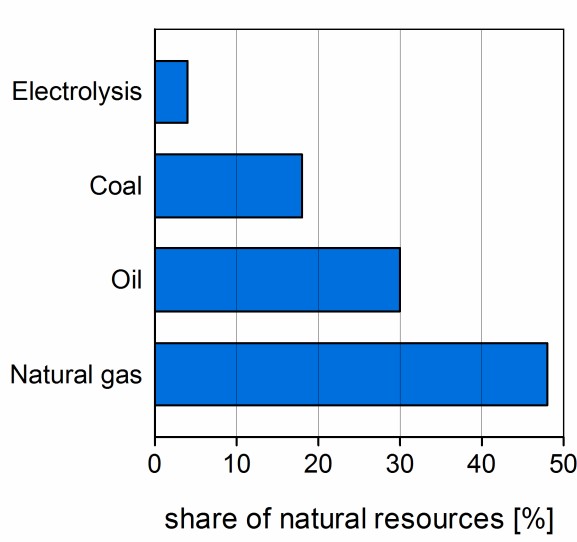

**Figure 4.** World hydrogen production.

When fossil fuels are used for hydrogen production, large amounts of pollution are generated, which is not environmentally friendly [36]. Therefore, there is a necessity to shift hydrogen production towards low-carbon hydrogen. Linking natural gas reforming (SMR) with CCUS technologies is the reasonable way to decarbonize the EU energy system by 2050 in comparison with the all-electric approach [37]. From 80 to 90% of $CO_2$ emissions can be removed using CCUS technologies [38]. The average carbon dioxide emission for natural gas reforming is about 9 kg $CO_{2eq}$/kg$H_2$. When natural gas reforming is integrated with CCS and 90% of carbon dioxide is removed, the average carbon dioxide emissions is 1 kg $CO_{2eq}$/kg $H_2$, respectively [39]. Investment costs of hydrogen production by the electrolysis of water are much higher than for hydrogen production from natural gas integrated with CCUS processes [40].

Current CCUS technologies allow for hydrogen production from natural gas at an industrial scale, which is especially important for economic sectors with hard-to-avoid emissions. There is a growing conviction in the EU that carbon capture, storage, and utilization technologies (CCUS) integrated with low-carbon hydrogen production will be an important element in the short-term transition towards a carbon-free economy [41,42]. There are today a few SMR projects coupled with CCUS under feasibility analysis: H-Vision in the Netherlands up to 0.6 Mt $H_2$/year and Magnum in the Netherlands with demand for 0.2 Mt $H_2$/year [43].

As an alternative technology for SMR, auto thermal reforming (ATR) is mentioned. ATR is cheaper as a result of more concentrated emissions and higher $CO_2$ recovery rates compared to SMR. ATR technology is already used in ammonia and methanol production, and there are new projects that plan to use that technology.

Hydrogen produced by the electrolysis of water covers only 4% of the world's hydrogen production. When the electricity is derived from renewable sources, the so-called power-to-hydrogen (P2H) concept of hydrogen production is called clean because greenhouse-gas emissions in this case are close to zero. Power-to-power (P2P) applications may use generated hydrogen for storage and reconversion into energy. The main developed electrolysis technologies today are alkaline electrolysis cells (AEC), proton exchange membrane electrolysis cells (PEMEC), and solid oxide electrolysis cells (SOEC). Taking into account the increasing environmental problems, renewable energy is the best alternative to conventional fossil fuels [44].

Unfortunately, the present hydrogen production covers only a small part of the global energy, which is still produced mainly from natural gas or coal [10], releasing only in the EU annually 70 to 100 million tons of $CO_2$. To reach the target of climate neutrality, the production of hydrogen needs to be fully decarbonized and reach a far larger scale.

It is estimated [43] that current hydrogen production, about 75 Mt $H_2$ each year, corresponds to more than 800 Mt $CO_2$ annually. The average reported emissions for natural gas and coal are, respectively, 9 t $CO_2$/t $H_2$ and 20 t $CO_2$/t $H_2$.

Today, there are seven projects in operation which generate hydrogen from fossil fuels integrated with CCUS, yielding over 0.4 Mt $H_2$ with the ability to capture about 6 Mt $CO_2$ [43]. Retrofitting existing facilities with CCUS significantly diminishes emissions from these facilities and enables them to have a continued sustainable operation. Combining CCUS technology with fossils-based hydrogen production is currently less expensive than water electrolysis using renewable energy.

The transition from renewables to a clean hydrogen strategy is expensive and requires the employment of costly end-use technologies [45] as well as the high costs involved with market deployment [46]. Therefore, efforts should be undertaken to reduce the costs of hydrogen production with low or zero carbon dioxide emissions and a cost-effective way of generating hydrogen energy to foster green hydrogen production on a large scale [47].

The reduction of hydrogen production costs and carbon dioxide emissions is investigated by a number of researchers. Valente et al. [48] have come to the conclusion that hydrogen production should be based on renewable energy sources to reduce its environmental impact. Haghi et al. [49] show the advantages of underground hydrogen storage, namely the reduction of hydrogen cost and emission of greenhouse gases. Im-orb et al. [50] confirmed that the production of hydrogen through water electrolysis is clean technology. Similarly Mehrpooya et al. [51] stated that the electrolysis of water combined with renewable energy generation may be considered the most environmentally friendly technology for hydrogen production.

Yadav and Banerjee [52] analyzed high-temperature steam electrolysis to produce hydrogen using solar energy. They found out that this method is not competitive and needs the reduction of component costs. El-Emam and Ozcan [42] propose to use nuclear and geothermal energy to obtain cheaper electricity and low-cost hydrogen. Generally, low-carbon hydrogen production embraces both green hydrogen produced from renewable electricity and blue hydrogen produced from fossil fuels with the use of CCUS technologies to remove $CO_2$ emissions. Green hydrogen as a clean technology is rapidly developing and is used worldwide in pilot and commercial-scale operations [53]. However, its costs are very high today, especially in comparison with fossil-based hydrogen [42,48]. In the case of blue hydrogen, the additional cost of CCUS technology is also high [54]. A promising new technology called aqua hydrogen was developed by Proton Technologies Canada Inc. This technology makes hydrogen production based on fossil fuels carbon free. It allows hydrogen extraction from conventional oil fields. What is important, in this case, is that fossil fuels may be used to produce zero-carbon hydrogen.

A comparative study on the environmental impact of hydrogen production methods depending on a renewable or a conventional kind of energy sources was presented in [55], and assessment of the life cycle for hydrogen production methods was discussed in [56].

Haris Ishaq et al. presented a detailed review concerning hydrogen production methods, storage, infrastructure, transportation, distribution, and utilization based on system design, costs, and efficiencies [17]. They studied natural gas reforming, coal gasification, water electrolysis, and the thermochemical Cu-Cl cycle.

A few large-scale projects for green hydrogen production are already running around the globe. Air Liquide [57] has the world's largest green hydrogen plant (20 MW) in Quebec, Canada, yielding daily up to 8.2 tons of green $H_2$. The OYSTER consortium is leading a project worth EUR 5 million investigating offshore hydrogen production [58].

Due to its properties and capabilities, hydrogen is a potential fuel option for electricity and transportation applications. It can be used as fuel in the transportation sector [59,60], as an energy carrier [61–63], and energy storage media [64–66]. Although hydrogen has shown its usefulness for clean-energy production, only a small fraction of currently produced hydrogen is applied for energy generation. The majority has been used as a feedstock for processing industries, with approximately 49% in ammonia production, 37% in petroleum refining, 8% in methanol production, and 6% in diverse smaller-volume uses. Balat [67] found out that hydrogen may be utilized as a fuel in combustion engines without any substantial modification. There are some advantages to hydrogen as a fuel for automobiles, such as rapid combustion speed, lack of toxic emissions, and high effective octane number [68].

In particular, hydrogen could be useful in heavy industrial sectors like the production of cement, iron and steel, chemicals, and synthetic fuels for ships and planes.

But, predominately, there is no sense to invert electricity from the grid to produce hydrogen, which then could be used in cars, homes, and commercial buildings. Electricity directly used for those needs is faster, easier, and cheaper.

Production of blue hydrogen from gas should prevent methane emissions as well as capture and sequester large quantities of carbon dioxide, for which currently, there is no significant capacity.

Blue hydrogen projects may prolong the life of the existing fossil-fuel infrastructure. New projects should meet certain social expectations and will require the acceptance and engagement of local communities.

### 3.2. Hydrogen Storage, Transportation, and Distribution

The hydrogen infrastructure compasses production, storage, transport, and distribution.

Hydrogen storage methods can be divided into two groups using physical and chemical processes (Figure 5). The main methods using physical processes are hydrogen compressing to pressures close to 700 atm, cryogenic liquefaction of hydrogen at temperatures of −253 °C, and indirect methods involving the compression of cooled hydrogen.

Chemical methods of storing hydrogen are based on adsorption and absorption processes. Adsorption methods include, for example, hydrogen adsorption on activated carbon grains, adsorption in carbon nanotubes, hydrogen adsorption in zeolites, and in complex structures: covalent organic frameworks (COFs), metal–organic frameworks (MOFs), and polymers of intrinsic microporosity (PIMs). Absorption methods involve the absorption of hydrogen in solutions, e.g., ammonia, ammonia borane, and liquid organic hydrogen carriers (LOHC), that can absorb and release hydrogen through chemical reactions and metal hydrides.

Hydrogen production, storage, and transportation costs are much higher in comparison to traditional energy sources. It is expected that hydrogen costs will decrease with time; nevertheless, the required investments to develop a hydrogen infrastructure are a major challenge [69].

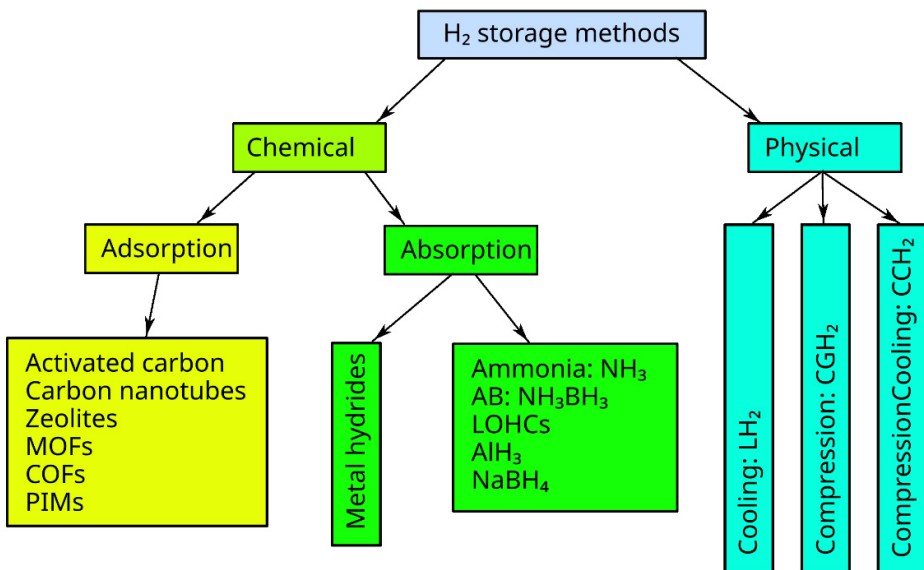

**Figure 5.** Hydrogen storage methods.

A comprehensive refuelling infrastructure system with fuelling stations in strategic areas is essential for the development of the sustainable global hydrogen market. On a small scale, such a system is already present for fuel-cell vehicles (FCVs), with more than 920 hydrogen stations globally reported at the end of 2023 (Figure 6).

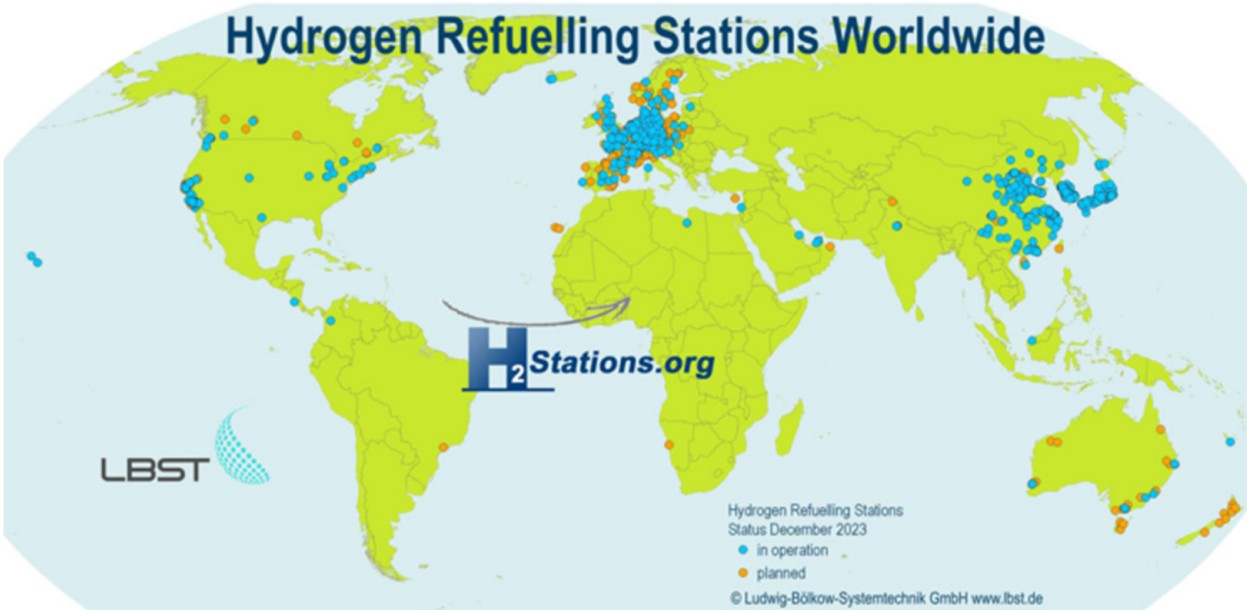

**Figure 6.** European conception of a hydrogen distribution network by Linde [70].

Figure 7 shows the planned European hydrogen distribution network (The European Hydrogen Backbone) [71]. Dark lines indicate existing pipelines that will be adapted to transport hydrogen, and yellow lines indicate new ones, dotted lines represent undersee pipelines and stars indicate cities. In a well-managed system with low leak rates, both green hydrogen and to a lesser extent blue hydrogen would significantly reduce warming compared with fossil fuels [72].

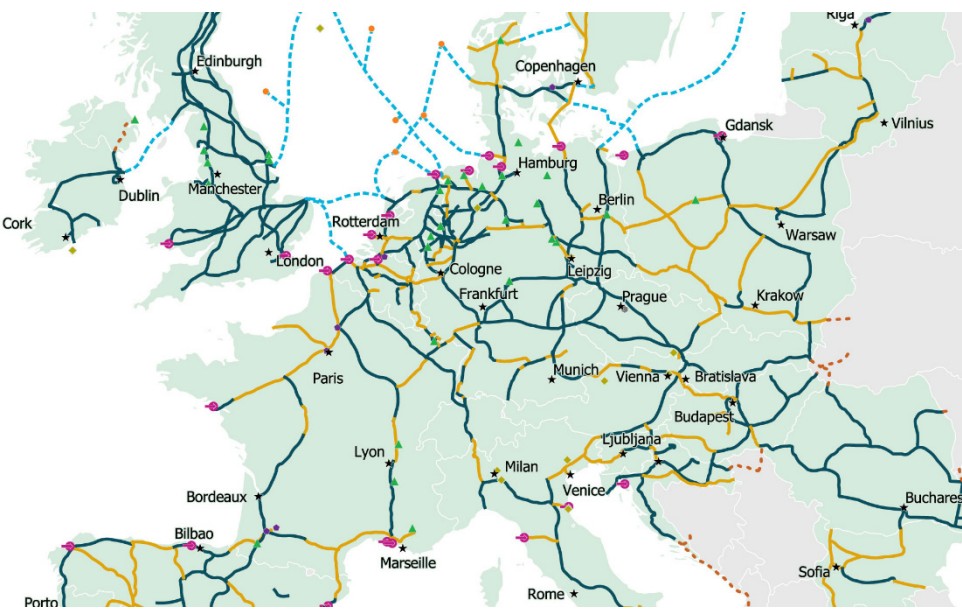

**Figure 7.** Conception of European hydrogen distribution network.

The technologies that enable the transport of commercially viable quantities of hydrogen in bulk are still in the developmental phases. Hydrogen is a challenging product to safely transport. It has the lowest density of all gases and is also highly flammable when mixed with any amount of air.

Contrary to common misconception LNG transport equipment cannot be easily adapted for hydrogen [73].

There are some promising recent technological developments in the transport of hydrogen by sea. Within a pilot project led by a consortium including Japan's J-Power, Kawasaki Heavy Industries (a major LNG tanker builder), Shell, and AGL, the first gas tanker capable of carrying liquefied hydrogen was built. There are also other pilot projects realized in South Korea (building a ship for carrying liquefied hydrogen in bulk, expected to be operational by 2027) and Norway (building specialized containers to carry liquified hydrogen and related ship cargo and distribution systems, expected to be operational by 2024 [64]).

Apart from that, new rules and regulations are prepared concerning the design and construction of ships, specialized containers to carry liquified hydrogen, and safe transportation of bulk hydrogen by road. However, standards for the transport of pure hydrogen in bulk are still in development.

## 4. Hydrogen and CCUS

EU policy assumes the production of green and blue hydrogen. Both renewable (green) hydrogen received from renewable sources of energy and decarbonized hydrogen (blue), received from natural gas coupled with CCUS techniques, are needed in long- and short-term conditions.

To achieve greenhouse-gas emission reduction according to the Paris Agreement goal, a significant change in the energy industry chain (production, storage, and consummation) will be needed in upcoming years. The Paris Agreement goal expects to hold the increase in the global average temperature to well below 2 °C above pre-industrial levels and pursues efforts to limit the temperature increase to 1.5 °C above pre-industrial levels. The proposed changes in practice lead to a net-zero emissions strategy, which requires that any anthropogenic $CO_2$ emissions be balanced by the removal of produced $CO_2$ through industrial or nature-based means, such as afforestation, reforestation, land-use changes, or the use of CCUS-based processes.

The International Energy Agency (IEA) report "Energy Technology Perspectives 2020" underlines the significance of CCUS technology in global energy system transformation. CCUS together with electrification based on renewable sources, bioenergy and hydrogen are considered as four pillars of the transformation process. As follows, carbon-removal and storage technologies may help to decrease emissions from large sources, power stations, or large industrial plants. By combining CCUS with bioenergy (BECCS) or by direct $CO_2$ capture from the air (DAC), it is possible to generate even negative emissions. It is pointed out that because of technical difficulties in removing emissions in certain sectors, such as steel, cement, chemicals, aviation, and transport, CCUS technologies will still be required. Therefore, CCUS technologies are strategic ones in the transition to net-zero emissions [9,43].

In Figure 8, hydrogen value-chain options are presented for low-carbon hydrogen combined with CCUS processes.

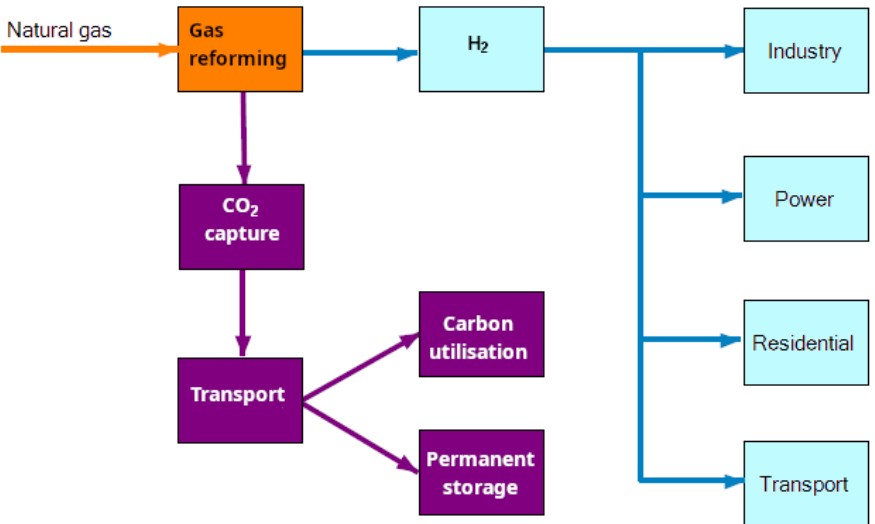

**Figure 8.** Hydrogen and CCUS value-chain options.

The application of CCUS in the production of hydrogen based on fossil fuels is the cheapest way today to produce low-carbon hydrogen. Not without significance is the fact that $CO_2$ capture technology can be adapted to existing or recently built plants and allow for their operation in the decades to come. Without CCUS, power and industrial plants existing today could be further emitting about eight billion tons of carbon dioxide in 2050 [9,43]. In heavy industrial sectors such as those producing cement, iron and steel, chemicals, and synthetic fuels, CCUS can help reduce emissions.

Today $CO_2$ from the majority of CCUS projects is used by oil companies in enhanced oil recovery (EOR), but $CO_2$ may be used to produce chemicals, synthetic fuels, or building materials.

CCUS technologies also offer the possibility to remove $CO_2$ from the atmosphere, DAC, or from processes of generating energy from biomass, BECCS technology (bioenergy combined with CCUS). Both BECCS and DAC represent so-called "negative emissions". These negative emissions can be used to balance emissions from sectors in which net-zero emission is not justified due to economic or technical reasons.

## 5. CCUS's Role in the Global Energy System

CCUS represents a strategic value for the transformation of the global energy system towards a net-zero goal by reducing existing emissions in particular economic sectors. In this way, CCUS provides a tool for the production of low-carbon hydrogen and the removal of carbon dioxide. When discussing $CO_2$ capture, different technologies are taken into account, such as carbon capture and storage (CCS), carbon capture and utilization (CCU), or carbon capture, utilization and storage (CCUS).

Therefore, progress in CCUS technologies, and their development and deployment, may have resulted in significant improvements for economic sectors, including cost reductions and infrastructure development.

It is estimated that, in 2019, 30% of global $CO_2$ emissions were emitted from coal power plants and that 60% of these power plants will be still in use in 2050, if not retired early. Thus, the only alternative solution for existing power and industrial plants, their infrastructure, and supply chains for continued use is integration with CCUS technologies. It can enable the preservation of production, employment, and economic prosperity and can help avoid the economic and social threats caused by early retirement. For example, Germany plans to retire 40 GW of coal-fired power plants before 2038, which will require a EUR 40 billion social package to compensate for the losses of coal mine and power plant owners and local communities [9].

In order to realize the net-zero goal, emissions from different economic sectors (energy, industry, and transport) must be reduced. This also includes sectors with hard-to-abate emissions like heavy industry sectors, which are accountable for about 20% of global $CO_2$ emissions [39], as well as aviation, road freight, and maritime shipping. Some sectors will simply not be able to reach net-zero emissions without coupling with CCUS. As an example, the production of cement is mentioned where limestone (calcium carbonate) is heated and large amounts of $CO_2$ are emitted. These emissions are not the result of fossil-fuel usage and are responsible for about 4% of all energy sector emissions. In this case, CCUS is the only option. In iron, steel, and chemicals production, CCUS can yield significant emissions reductions. Currently, in the production of virgin steel, fertilizers and methanol low-carbon hydrogen based on CCUS are the cheapest solutions for reducing emissions [39].

The IEA report states that CCUS is the only solution to reduce $CO_2$ emissions from natural gas processing. Natural gas layers may contain even up to 90% of $CO_2$, which needs to be removed before the gas can be sold or liquefied. On the other hand, this $CO_2$ can be stored in geological formations or used in the EOR process instead of releasing it into the atmosphere.

CCUS is also an option to decarbonize long-distance transport, including aviation. As an alternative to fossil fuels for aviation, synthetic hydrocarbon fuels and biofuels based on $CO_2$ supply can be used. The required $CO_2$ in this case should come from bioenergy or from the air to realize the net-zero emission goal.

Hydrogen is considered a universal energy carrier that is able to support the decarbonization of different economic sectors [39], even those with hard-to-abate emissions. CCUS can help decrease emissions from the already existing production of hydrogen, which is based on natural gas and coal (grey hydrogen) methods, which are accountable for more than 800 Mt $CO_2$ annually.

Carbon-removal technologies such as CCUS, BECCS, and DAC, together with natural processes like afforestation, reforestation, or natural bioprocesses, can help reduce or even achieve some net negative emissions in heavy industry sectors or technically challenging and prohibitively expensive sectors.

Although carbon capture and storage (CCS) is a feasible method, there are still various challenges to deploying this technology concerning social resistance and environmental concerns. In these cases, carbon capture and utilization (CCU) technologies can be favourable [74], and some additional costs tied with CCS can be avoided, for example, $CO_2$ purification, additional infrastructure expense, and storage and monitoring costs [75–77]. Currently, the permanent CCU storage technologies include carbonation, mineralization, and EOR. Synthetic fuels and chemicals are temporary storage media because, when used, the $CO_2$ contained in them is again released into the atmosphere [78,79]

CCUS offers a way to decarbonize the production of low-carbon hydrogen from existing hydrogen plants by decreasing emissions and may provide the least cost-efficient solution for scaling up new hydrogen production [43].

## 6. Hydrogen Production Costs

In Figure 9, the average costs of hydrogen production are presented. As can be seen, the costs of hydrogen from fossil fuels are about EUR 0.5–2/kg and are significantly lower than costs for electrolysis based on renewable energy, EUR 3–7.5/kg [39].

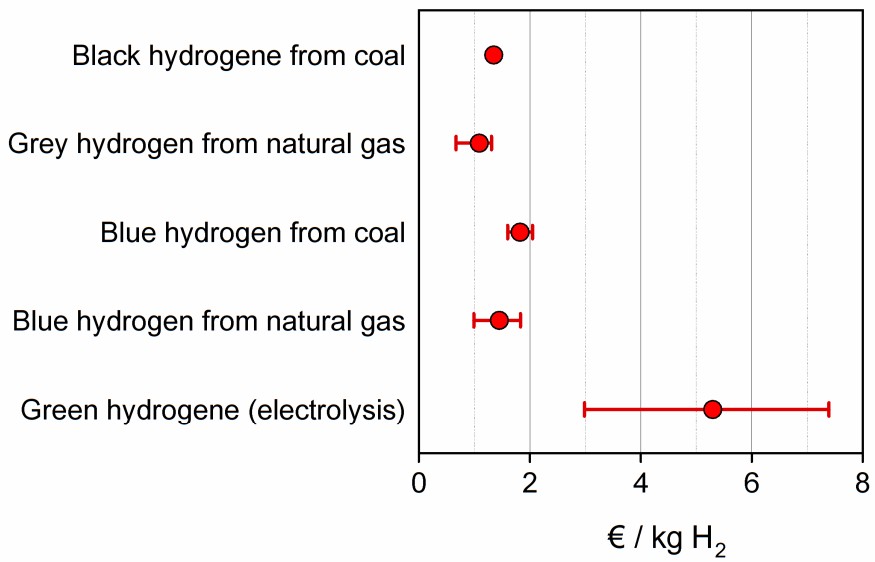

**Figure 9.** Hydrogen production costs.

Presently, hydrogen based on fossil fuels is still cost competitive when compared to hydrogen based on renewable energy or fossil hydrogen produced from fossil fuels integrated with CCUS processes.

A comparison of hydrogen production costs for different production methods and energy sources (with gasoline for reference) was presented in Table 3 [80].

**Table 3.** Cost of hydrogen based on various production processes.

| Process | Cost of Hydrogen (€ per kg) |
| --- | --- |
| Natural gas reforming | 0.96 |
| Natural gas + $CO_2$ capture | 1.13 |
| Coal gasification | 0.89 |
| Coal + $CO_2$ capture | 0.96 |
| Wind electrolysis | 6.18 |
| Biomass gasification | 4.31 |
| Biomass pyrolysis | 3.53 |
| Nuclear thermal splitting of water | 1.52 |
| Gasoline (for reference) | 0.86 |

In comparison with the gasoline price (as a reference), the current price of hydrogen from renewable energy sources is very high.

A summary of various hydrogen production technologies with their efficiencies is presented in Table 4 [80].

**Table 4.** Hydrogen production technologies summary.

| Technology | Feedstock | Efficiency | Maturity |
|---|---|---|---|
| Steam reforming | Hydrocarbons | 70–85% | Commercial |
| Auto-thermal reforming | Hydrocarbons | 60–75% | Near term |
| Biomass gasification | Biomass | 35–50% | Commercial |
| Electrolysis | $H_2O$ + electricity | 50–70% | Commercial |
| Photolysis | $H_2O$ + sunlight | 0.5% | Long term |
| Thermochemical water splitting | $H_2O$ + heat | NA | Long term |

As can be seen, only steam reforming, biomass gasification, and electrolysis achieved currently the commercial maturity stage. The other technologies need more research and development. Only steam reforming efficiency exceeds 80%.

It is estimated that with carbon dioxide emissions prices in the range of EUR 55–90/tons of $CO_2$, gas- or coal-based hydrogen with CCUS (blue hydrogen) costs will be lower than for gas- or coal-based hydrogen without CCUS (black or grey hydrogen).

Green hydrogen costs are supposed to be going down. Electrolyser costs are expected to decrease by about 50% by 2030 and may be competitive with fossil-based hydrogen costs, especially in regions with cheap renewable electricity. The fossil-based hydrogen and renewable hydrogen costs in the EU in 2030 were predicted to be, respectively, between EUR 2–2.5/kg and EUR 1.1–2.4/kg [39]. But taking into account the currently unstable and rising prices of energy sources, these forecasts are uncertain and should be considered carefully.

To minimize the negative impact of the hydrogen sector on the environment and climate, a life-cycle study approach will be needed. An appropriate support and trade policy will be expected in a transition phase to realize the European Green Deal and EU Strategy for Energy System Integration tenets [2,5] based on low-carbon hydrogen. These strategies contribute towards and are in line with the goals of sustainable development and the tenets of the Paris Agreement.

## 7. Pros and Cons of Low-Carbon Hydrogen Production

Currently, green hydrogen is produced from renewable sources, blue hydrogen is based on fossil-fuel sources with CCUS technology, and white/purple/aqua hydrogen is based on the natural form of $H_2$ in underground deposits, in oceans, and the air, as well as hydrogen based on nuclear power and new technologies, may be considered as parts of the low-carbon economy. All of them represent some challenges that may affect future directions and recommendations.

The low-carbon hydrogen production costs are the base of the hydrogen economy. Unfortunately, current green hydrogen technology with reasonable efficiency is not available and production costs are too high [81,82]. The price of green hydrogen produced from renewable electricity is in the range of EUR 2.28–7.43/kg and is much higher than black and grey hydrogen and higher than blue hydrogen (Figure 8). Green hydrogen costs are too high for a wide deployment; it is estimated that the price of green hydrogen may fall low enough around the year 2030 [83,84].

There are mentioned two main factors influencing the price of green hydrogen. These are the cost of electrolysis and the price of green electricity. Presently, the capacity of global electrolysis is limited and green electricity prices are too high. Thus, water electrolysis is related to high costs [85] and large energy demands [86].

To reduce the green hydrogen cost in the long term, the cost of renewable electricity should be lowered [87] and some future technological innovations introduced to achieve the maturity of electrolysis-based technologies [42].

The growing number of wind and solar power generation systems has a significant effect on renewable [88] electricity and large-scale green hydrogen application. However,

the high capital cost of a renewable energy infrastructure and its weather-dependent character and intermittency may increase the cost of generated green hydrogen [42].

The costs of blue hydrogen are higher in comparison with black hydrogen and grey hydrogen, Figure 8. The costs of blue hydrogen mainly depend on the coal and natural gas prices. But not without significance is the cost of capture, storage, and utilization of carbon dioxide.

According to the Global Carbon Capture and Storage Institute, there were 18 large CCS projects running worldwide in 2018. But at present, CCUS technologies are still immature, with high energy consumption, with the need for large-scale applications, and with a weak transport link. Generally, the CCUS technologies are in the early stages of development with high costs and low efficiency.

Currently, the capital cost of a blue hydrogen plant coupled with CCUS technologies is about half that of an electrolytic plant. But in the future, with supposed falling costs of renewable energy and rising prices of carbon emission, the costs of green hydrogen will decrease and make blue hydrogen less competitive [89].

Carbon dioxide emissions in the case of blue hydrogen are lower than in the case of black and grey hydrogen, but there are still some emissions that cannot be avoided. As it is estimated, there is 5–15% leakage. The $CO_2$ capture for the SMR and ATR processes is 85–90% and 90%, respectively. Thus, large-scale blue hydrogen production will release millions of tons of emissions each year which may cause potential environmental uncertainty.

Aqua hydrogen is quite a new technology which needs development, scaling up, an increase of investment, and promotion. Aqua hydrogen is produced underground, at a temperature above 350 °C, and $CO_2$ is stored underground. But, some environmental concerns and hazards are noticed, such as the increased water acidity, infringement of ecological balance, $CO_2$ leakage, impact on health, and social acceptance.

## 8. Trends, Challenges, and Limitations in Hydrogen Technology

As a clean-energy carrier, hydrogen is supposed to play a significant role in the implementation of 2050 net-zero targets.

Over the last decade, the global decarbonization efforts and developments in existing technologies accelerated the main hydrogen trends impacting the hydrogen economy in 2024. In the presented industry research [90], the biggest impact was found for hydrogen fuel cells currently contributing to greenhouse-gas emissions reduction in zero-emissions heavy-duty vehicles. Fuel cells find applications in marine, land, and aviation operations, including ships, trains, planes, drones, cars, trucks, buses, and heavy industrial vehicles. The Australian startup H2X designs and develops hydrogen fuel-cell-powered vehicles. US-based startup BWR Innovations provides portable hydrogen fuel-cell solutions.

The diversity and accessibility of renewable energy systems, as well as advanced electrolysis technologies (AEM Electrolyzers, Large-Scale Electrolyzers), allow for sustainable hydrogen production from renewable energy sources.

Using renewable energy systems for green hydrogen eliminates carbon emissions common in conventional hydrogen production. Other options for hydrogen production using solar energy include photocatalytic and thermochemical water splitting.

Renewable hydrogen and CCUS have a major influence because of their inter-relationship with the production of clean hydrogen.

Other sustainable hydrogen production methods are biomass gasification and the set of X-to-Hydrogen-to-X technologies. Hydrogen distribution and storage are closely linked, as distribution depends on the fuel's proper storage and handling capabilities. Combined heat and power and green propulsion represent important applications of hydrogen which show the availability of hydrogen as an energy carrier.

Generally, the following trends and their impact on the hydrogen economy in 2024, include hydrogen fuel cells (24%), renewable hydrogen (15%), advanced electrolysis (15%), x-to-hydrogen-to-x (19%), hydrogen carriers (9%), carbon capture (8%), utilization and

storage (24%), hydrogen distribution (6%), hydrogen liquefaction and compression (6%), combined heat and power (5%), and hydrogen propulsion (1%) [90].

Unfortunately, the current state of hydrogen infrastructure is not sufficient. The shortcomings of production, storage, transportation, and distribution facilities challenge the adoption of hydrogen energy as a mainstream energy source in the future [91]. There are also obstacles being reported concerning hydrogen strategy. These are challenges related to hydrogen deployment and the costs of renewable hydrogen, which are too high in comparison with hydrogen based on fossil fuels. Further investments, regulatory frameworks, new markets, research, innovations, and a large-scale infrastructure network are needed.

The development of hydrogen infrastructure is associated with significant challenges [72]. The infrastructure will have to operate at high pressures to store and transport hydrogen. It can be expensive and technically difficult. For transportation, hydrogen needs to be compressed to over 700 atm. It will require specialized compressors and storage containers.

The current technology is not yet efficient enough and not ready to be used for large-scale hydrogen production or deployment. Other critical issues that must also be addressed are durability and leaking. The infrastructure's durability is significant for cost reductions and end-user safety. Hydrogen is a highly flammable gas that can leak from pipelines or storage tanks, creating safety concerns and increasing maintenance costs.

## 9. Conclusions

The increasing continued use of fossil fuels accompanied by growing greenhouse-gas emissions leads to environmental problems and air pollution. To reach climate neutrality by 2050 and reach the Paris Agreement target of holding the increase of the global temperature, deep decarbonization and reduction of greenhouse-gas emissions are required.

In the research on fossil-fuel alternatives, hydrogen was recognized as a clean carbon-free solution. Hydrogen is an efficient energy carrier and storage media and can be used to decarbonize many sectors of the global economy.

In this study, the hydrogen energy industry chain was described. The production methods, storage methods, distribution infrastructure network, and hydrogen applications were analyzed. The building and development of a hydrogen-based economy needs an investigation and analysis of the full hydrogen energy industry chain. Renewable and low-carbon sources should be defined. The hydrogen production efficiency, as well as a relevant infrastructure to supply hydrogen to the end users, should be determined. A hydrogen distribution network should be developed and presented together with planned hydrogen hub centres. The market for increasing hydrogen supplies and demands must be created by decreasing the costs of new clean technologies and determining affordable energy costs from renewable sources. A life-cycle analysis and further study are also required to avoid the negative impacts of the hydrogen sector on climate and the natural environment. Further research on the hydrogen energy industry chain is still needed for hydrogen to become a key for the carbon-neutral global energy system and the new hydrogen economy.

Comprehensive information on hydrogen production, storage, transportation infrastructure performance, and stationary hydrogen applications are provided and compared. The new trends and technological developments for clean hydrogen production, storage and utilization are presented in the study.

The EU policy Strategy for Energy System Integration and EU Hydrogen Strategy ultimately insists on the production and development of renewable hydrogen (green hydrogen) and hydrogen produced from fossil fuels coupled with CCUS technologies (blue hydrogen) in the short-term conditions to decrease emissions and develop a manageable market.

Renewable energy sources and CCUS technologies are supposed to play a main role in diminishing and reducing greenhouse-gas emissions. The IEA reports that, by 2040, solar energy and wind power will become the largest source of low-carbon electricity. In the European Union, renewables will account for 80% of new capacity soon after 2030 [1,92].

It should be noted that electricity generation based on renewable sources is weather dependent, which results in instability and potential problems in energy supply and demand. To manage the imbalance of these sources, hydrogen is more likely to be the most promising solution for energy storage from renewable sources [4,93]. Thus, hydrogen is considered today as a primary technology for the long-term storage of renewable energy.

Green hydrogen is the main route for carbon-free energy production. Therefore, many countries make an attempt to develop green hydrogen production from renewable sources. However green hydrogen costs are presently too high. So, in the short term, fossil-fuel hydrogen will be used. According to the European Union reports, this transition period may take a decade or more [94]. Blue hydrogen is to be used in a transition stage to help achieve carbon-free hydrogen production in the decades to come.

There is a need to foster green hydrogen production on a large scale to diminish its costs more quickly. The green hydrogen development needs the lowering of electrolysis costs and green electricity prices, governmental policy frameworks [95], and public acceptance.

Blue hydrogen based on fossil fuels together with CCUS technology can reduce emissions of greenhouse gases at a relatively low cost. However, there are also potential environmental and technical challenges using CCUS [96], such as risks of $CO_2$ leakage, environmental impact, limited $CO_2$ capture efficiency, and the high cost of the CCUS technology. There is still a need to increase research and government support for CCUS technology.

New technologies such as aqua hydrogen based on fossil fuels may allow for obtaining carbon-free hydrogen at a low cost. But as a new technology, it needs commercial investment and promotion, especially for large-scale production. The environmental concerns related to this technology require a full life-cycle approach assessment of the environmental impact.

There are also obstacles being reported for the application of hydrogen in industrial processes and transportation such as costs of the green and blue hydrogen which are not competitive today and also the high costs of necessary investments into hydrogen equipment, storage, and utilization facilities. Other problems include deployment challenges, scaling up, connecting renewable energy production capacity to the electrolysers, and retrofitting the existing plants with CCUS technology. Further investments, regulatory frameworks, new markets, research, innovations, and a large-scale infrastructure network are needed.

CCUS technologies represent strategic value in the transition process to net-zero emission. CCUS can favour low hydrogen production from natural gas or coal and provide low-carbon hydrogen at a lower cost in the near future. Currently, the cost of hydrogen production integrated with CCUS is at least 50% lower than hydrogen production based on electrolysis and renewable sources of energy. It is estimated that CCUS coupled with hydrogen production would be a competitive solution, even with the declining costs of electrolysers and renewable electricity.

The application of CCUS in the production of hydrogen based on fossil fuels is the cheapest way to produce low-carbon hydrogen today. CCUS can help decrease emissions from the already existing production of hydrogen, which is based on natural gas and coal (grey hydrogen) and is responsible for more than 800 Mt $CO_2$ annually. Of great importance is also the fact that CCUS can be coupled with existing power and industrial plants, which without CCUS could emit about eight billion tons of carbon dioxide in 2050. CCUS can help reduce carbon dioxide emissions in heavy industrial sectors, such as the production of cement, iron and steel, chemicals and synthetic fuels, or in aviation.

CCUS can also remove $CO_2$ from the atmosphere to manage emissions that are unavoidable or technologically difficult to remove. Although CCUS technologies are useful, challenges exist in the deployment of these technologies taking into account social resistance and environmental concerns.

In a transition stage toward a clean hydrogen economy, appropriate support will be needed for low-carbon hydrogen to realize the targets of the European Green Deal and the

Strategy for Energy System Integration [2,39], which contribute towards the achievement of the Sustainable Development Goals and the tenets of the Paris Agreement.

**Funding:** This research received no external funding.

**Conflicts of Interest:** The authors declare no conflict of interest.

## Abbreviations

| | |
|---|---|
| AEC | Alkaline Electrolysis Cells |
| ATR | Auto Thermal Reforming |
| BECCS | Bio-Energy with Carbon-Capture and Storage |
| CCS | Carbon Capture and Storage |
| CCU | Carbon Capture and Use |
| CCUS | Carbon Capture Utilization and Storage |
| COFs | Covalent Organic Frameworks |
| DAC | Direct $CO_2$ Capture From Air |
| EOR | Enhanced Oil Recovery |
| EU | European Union |
| HHV | Higher Heating Value |
| IEA | International Energy Agency |
| LHV | Lower Heating Value |
| LOHC | Liquid Organic Hydrogen Carriers |
| MOFs | Metal–Organic Frameworks |
| OTEC | Ocean Thermal Energy Conversion |
| P2H | Power to Hydrogen |
| P2P | Power to Power |
| PEMEC | Proton Exchange Membrane Electrolysis Cells |
| PIMs | Polymers of Intrinsic Microporosity |
| SMR | Steam Methane Reforming |
| SOEC | Solid Oxide Electrolysis Cells |

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
