# Peer review of "Assessment of Hydrogen Energy Industry Chain Based on Hydrogen Production Methods, Storage, and Utilization"

_energies, doi:10.3390/en17081808_

Round 1

Reviewer 1 Report

Comments and Suggestions for Authors

The paper titled "Assessment of hydrogen energy industry chain based on hydrogen production methods, storage, and utilization" submitted to Energies delivers a comprehensive overview of the hydrogen energy industry chain, focusing on production techniques, storage, transportation, and utilization to achieve climate neutrality by 2050. The manuscript critically examines the costs, future prospects, and environmental impact of different hydrogen production methods. It discusses the role of hydrogen in decarbonizing various sectors and the challenges and opportunities within the hydrogen economy, offering insights into the development of a sustainable and integrated energy system.

However, potential improvements could include directly comparing hydrogen's efficiency and sustainability with those of other energy sources, providing detailed case studies demonstrating real-world applications, and expanding on the socio-economic impacts of transitioning to a hydrogen-based economy. Furthermore, exploring the technological and logistical challenges in scaling up hydrogen infrastructure would also add to the paper's comprehensiveness. 

Therefore these few areas could be improved upon in the manuscript:

1) Comparative analyses with other energy sources would provide a clearer picture of hydrogen's role and potential within the broader energy landscape.

2) Expanding the discussion on the economic and environmental implications of transitioning to a pure hydrogen-based energy system, as well as any potential disadvantages, that could bring.

Overall, the manuscript makes a significant contribution to the literature on hydrogen energy, offering insightful analyses and future directions for research and policy. With the suggested revisions, it would serve as a valuable resource for researchers, policymakers, and industry stakeholders interested in the hydrogen energy industry chain.

Author Response

The answer to Reviewer #1

The authors would like to thank Reviewer for precise and very helpful review of their article. All comments were taken into account, answered and corrected.

  1. Additional information regarding the comparison of the costs of hydrogen production according to different energy sources and technologies is presented in Tables 3 (Cost of hydrogen based on various production processes) and Table 4. Hydrogen production technologies summary) was added.
  2. More supporting information and literature references regarding hydrogen production and utilization was added in chapter 3.1 “Hydrogen production and utilization” (lines172-176, 209-212, 219-222, 289-310). Analysis of potentials, challenges and limitations of hydrogen technology was presented in new added Chapter 8 (lines 558-603).

Reviewer 2 Report

Comments and Suggestions for Authors

This work titled " Assessment of hydrogen energy industry chain based on hydrogen production methods, storage and utilization" is very important topic, but the paper needs a serious major revision as the review has not given new topics differs from previous reviews done in the same direction:

1- The Abstract must be more focused on the given topic

2- Introduction part needs more supporting information and literature as well about all aspect regarding hydrogen production and utilizations.

3- The author has nor given the sufficient information about the hydrogen trend in many applications and also there is no comparisons made between hydrogen economical value and other types of fuel.

4- There is an important thing needs to be made about SWOT analysis and also the author must talk about limitations, challenges and potentials as well related to hydrogen technology.

6- There are so many figures are very poor in quality and needs to be higher in resolution as well. 

Comments on the Quality of English Language

There are some minor modifications needs to be done and the manuscript needs to be checked again for these typo errors 

Author Response

The answer to Reviewer #2

The authors would like to thank Reviewer for precise and very helpful review of their article. All comments were taken into account, answered and corrected:

  1. The Abstract was corrected and focused more on the topic of the article.
  2. More supporting information and literature references regarding hydrogen production and utilization was added in chapter 3.1 “Hydrogen production and utilization” (lines172-176, 209-212, 219-222, 289-310).
  3. New chapter was added: Chapter 8. “Trends, challenges and limitations in hydrogen technology”. To compare hydrogen economical value Table 3 (Cost of hydrogen based on various production processes) and Table 4 (Hydrogen production technologies summary) were added (lines 480-493).
  4. Analysis of potentials, challenges and limitations of hydrogen technology was presented in added Chapter 8 (lines 558-603).
  5. The figures were prepared again with higher resolution. Fig.7 was changed due to low resolution.
  6. Manuscript was checked for typo errors.

Round 2

Reviewer 2 Report

Comments and Suggestions for Authors

The author have made the necessary corrections and now it is suitable for publications,